# Observation of Mermin-Wagner behavior in LaFeO₃/SrTiO₃ superlattices

M. Kiaba ®[1] ✉, A. Suter ®[2], Z. Salman ®[2], T. Prokscha ®[2], B. Chen ®[3], G. Koster ®[4] & A. Dubroka ®[1,5]

Two-dimensional magnetic materials can exhibit new magnetic properties due to the enhanced spin fluctuations that arise in reduced dimension. However, the suppression of the long-range magnetic order in two dimensions due to long-wavelength spin fluctuations, as suggested by the Mermin-Wagner theorem, has been questioned for finite-size laboratory samples. Here we study the magnetic properties of a dimensional crossover in superlattices composed of the antiferromagnetic LaFeO₃ and SrTiO₃ that, thanks to their large lateral size, allowed examination using a sensitive magnetic probe − muon spin rotation spectroscopy. We show that the iron electronic moments in superlattices with 3 and 2 monolayers of LaFeO₃ exhibit a static antiferromagnetic order. In contrast, in the superlattices with single LaFeO₃ monolayer, the moments do not order and fluctuate to the lowest measured temperature as expected from the Mermin-Wagner theorem. Our work shows how dimensionality can be used to tune the magnetic properties of ultrathin films.

The properties of magnetic films with thickness in the nanoscale have been a long-standing research topic. The theory of critical behavior predicts that the phase transition temperature should decrease with decreasing film thickness[1], which was observed in several cases[2–7]. In the 2-dimensional (2D) limit, Mermin and Wagner[8] extended the initial idea of Hohenberg[9] for a superconductor and predicted complete suppression of the long-range magnetic order in models with continuous rotational symmetries (i.e., with the Heisenberg or $XY$ spin Hamiltonian) at finite temperature due to long-wavelength fluctuations. Importantly, this prediction is strictly valid only for the thermodynamic limit, i.e., for samples with laterally infinite sizes. However, since the divergence of the fluctuations in the 2D case is only slow (logarithmic in sample size), it was suggested that for any finite-size laboratory samples, the phase order is preserved for superconductivity[10] and even for magnetism[11].

The discovery of magnetic van der Waals materials allowed the investigation of magnetism in samples with thickness down to a single monolayer[12]. For example, it was reported that in samples of bulk antiferromagnet NiPS₃ that are two or more monolayers thick, the magnetic order is preserved, whereas it is suppressed in a single monolayer sample[13]. Since the Hamiltonian of NiPS₃ has the $XY$ symmetry, this behavior thus follows the prediction of the Mermin−Wagner theorem rather than the suggestions for preserving the long-range order[11]. However, due to the small lateral size of the single monolayer NiPS₃ samples obtained by the exfoliation, the antiferromagnetic order was probed relatively indirectly by Raman spectroscopy via coupling of a phonon to a magnon mode[13].

To test the Mermin−Wagner behavior using a magnetic probe, we study the magnetic properties of three to two-dimensional crossover in superlattices composed of antiferromagnetic LaFeO₃ separated by nonmagnetic SrTiO₃ layers. Bulk LaFeO₃ is a prototypical perovskite antiferromagnetic insulator with Heisenberg symmetry of the spin Hamiltonian[14] and with the highest Néel temperature ($T_N$) of 740 K among $Re$FeO₃ materials[15], where $Re$ stands for rare earth. It has a high magnetic moment of almost 5 $\mu_B$ per Fe³⁺ ion and the G-type structure of the antiferromagnetic state (where each spin is aligned opposite to the nearest neighbor). Thus the antiferromagnetic order is expected to be relatively robust. Thanks to the advancement in deposition technology, it is possible to fabricate heterostructures with sharp interfaces that are

¹Department of Condensed Matter Physics, Faculty of Science, Masaryk University, Kotlářská 2, 611 37 Brno, Czech Republic. ²Laboratory for Muon-Spin Spectroscopy, Paul Scherrer Institute, 5232 Villigen PSI, Switzerland. ³Key Laboratory of Polar Materials and Devices (MOE) and Department of Electronics, East China Normal University, 200241 Shanghai, China. ⁴MESA+ Institute for Nanotechnology, University of Twente, 7500 AE Enschede, The Netherlands. ⁵Central European Institute of Technology, Brno University of Technology, 612 00 Brno, Czech Republic. ✉e-mail: kiaba@mail.muni.cz

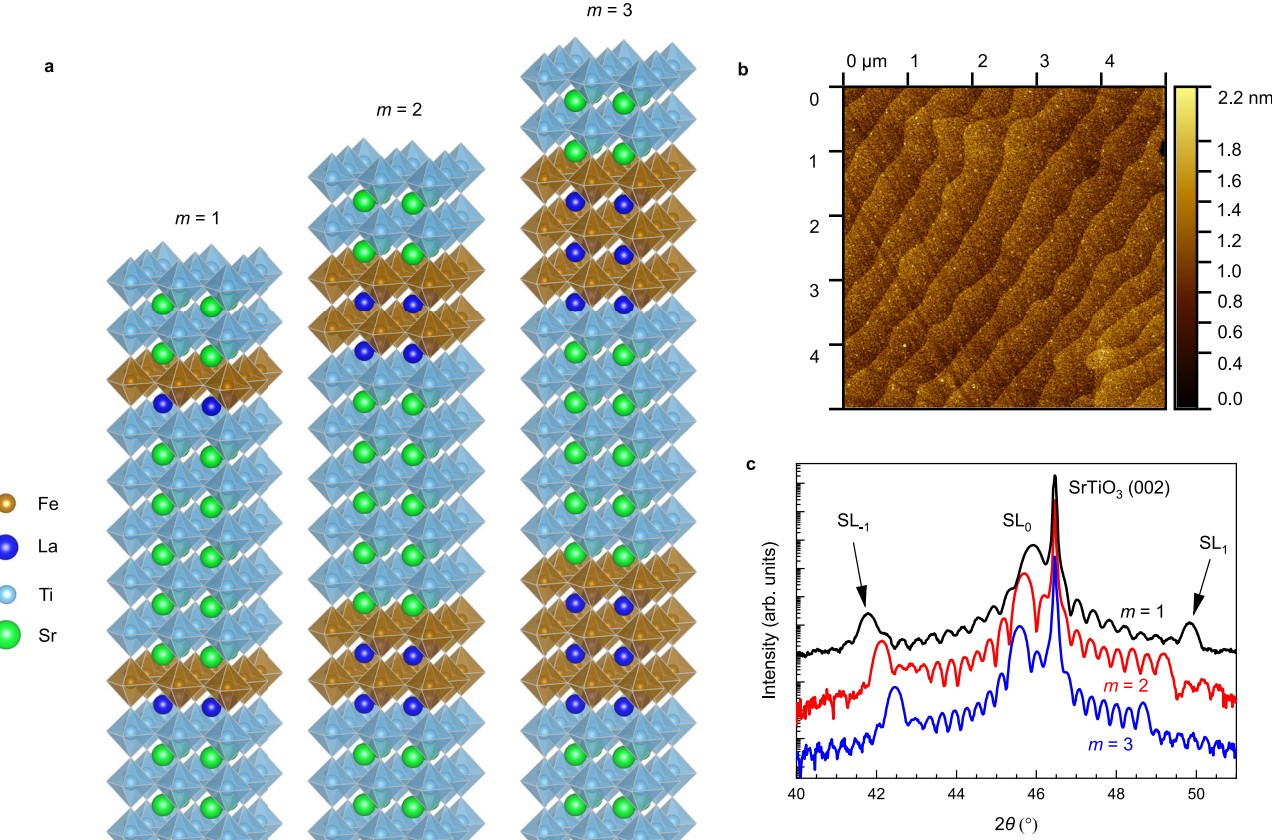

**Fig. 1 | Structural characterization of the superlattices. a** Scheme of [(LaFeO₃)$_m$/(SrTiO₃)₅]₁₀ superlattices near the surface of TiO-terminated SrTiO₃ (001) substrate. **b** The surface morphology of the $m = 2$ superlattice determined by an atomic force microscope. **c** X-ray diffraction spectra exhibiting zero (SL₀) and the first (SL₁, SL₋₁) diffraction peaks due to the (LaFeO₃)$_m$/(SrTiO₃)₅ bilayer occurring near the (002) diffraction of the SrTiO₃ substrate.

composed of perovskite oxides with various order parameters, including magnetism, ferroelectricity, and superconductivity[16,17]. Perovskite oxide heterostructures are also promising for applications since they can be used in large-scale samples and devices (see, e.g., refs. [18–21]). Using pulsed laser deposition, we fabricated superlattices with 1–3 monolayers of LaFeO₃ separated by a non-magnetic spacer of 5 monolayers of SrTiO₃ with a large lateral size of $10 \times 10$ mm² that allowed their investigation using a sensitive magnetic probe−low-energy muon spin rotation spectroscopy[22].

To enhance the signal in the muon spin rotation experiment, we prepared superlattices denoted as [(LaFeO₃)$_m$/(SrTiO₃)₃]₁₀, where a bilayer with $m = 1, 2$ or 3 monolayers of LaFeO₃ and five monolayers of SrTiO₃ is repeated 10 times. The scheme of the ideal superlattice structure near the interface with the TiO-terminated SrTiO₃ (001) substrate is shown in Fig. 1a. Figure 1b displays the surface morphology of the $m = 2$ superlattice measured by an atomic force microscope, which exhibits a flat surface with single unit cell steps similar to those of the substrate. The X-ray diffraction spectra (see Fig. 1c), exhibit zero (SL₀) and the first (SL₁, SL₋₁) superlattice diffraction peaks due to the (LaFeO₃)$_m$/(SrTiO₃)₅ bilayer, which depict that the superlattices have high structural quality with a negligible or low level of ionic diffusion. The thickness of (LaFeO₃)$_m$/(SrTiO₃)₅ bilayer determined from the first order diffraction peak follows very well the estimates based on the lattice constant of SrTiO₃ and LaFeO₃ (see Supplementary Fig. 1a).

Investigations of magnetic properties of ultrathin antiferromagnetic layers is a challenging task because of their zero (or very small) average magnetic moment compared to the large total diamagnetic moment of the substrate. To probe the magnetic properties of our superlattices, we have used muon spin rotation spectroscopy,

which is sensitive to even very weak local magnetic fields and can distinguish between static and dynamic behavior. We performed the experiments with a low-energy (2 keV) muon beam[22,23], where spin-polarized muons are implanted into the sample only within about 25 nm from the surface (see Supplementary Fig. 4). Any magnetic field component transverse to the muon spin direction causes its precession with the Lamour frequency $\omega_L = \gamma_\mu B$, where $\gamma_\mu$ is the gyromagnetic ratio of the muon and $B$ is the magnitude of the local magnetic field. The time dependence of polarization of the muon spin ensemble (the so-called asymmetry) is measured thanks to the muon decay into a positron preferentially emitted in the direction of the muon spin[24].

## Results

### Zero-field muon spin rotation

Figure 2 shows results from the muon spin rotation experiment in zero magnetic field. The time dependence of the muon spin polarization of the superlattices with $m = 3$ and 2 (see Fig. 2a and b, respectively), exhibit at high temperature a concave Gaussian-like profile and a transition to an exponential-like relaxation at lower temperatures. In contrast, the asymmetry of the $m = 1$ superlattice shown in Fig. 2c is qualitatively different because it exhibits a convex profile and a relatively slower relaxation rate. To get a more quantitative insight, we analyzed the zero field asymmetry, $A_{ZF}(t)$, with the phenomenological stretched exponential function[25–29]

$$A_{ZF}(t) = A_0 e^{-(\lambda t)^\beta},\tag{1}$$

where $A_0$ is the initial asymmetry, $\lambda$ is the depolarization rate, $\beta$ is the stretching exponent, and $t$ is time.

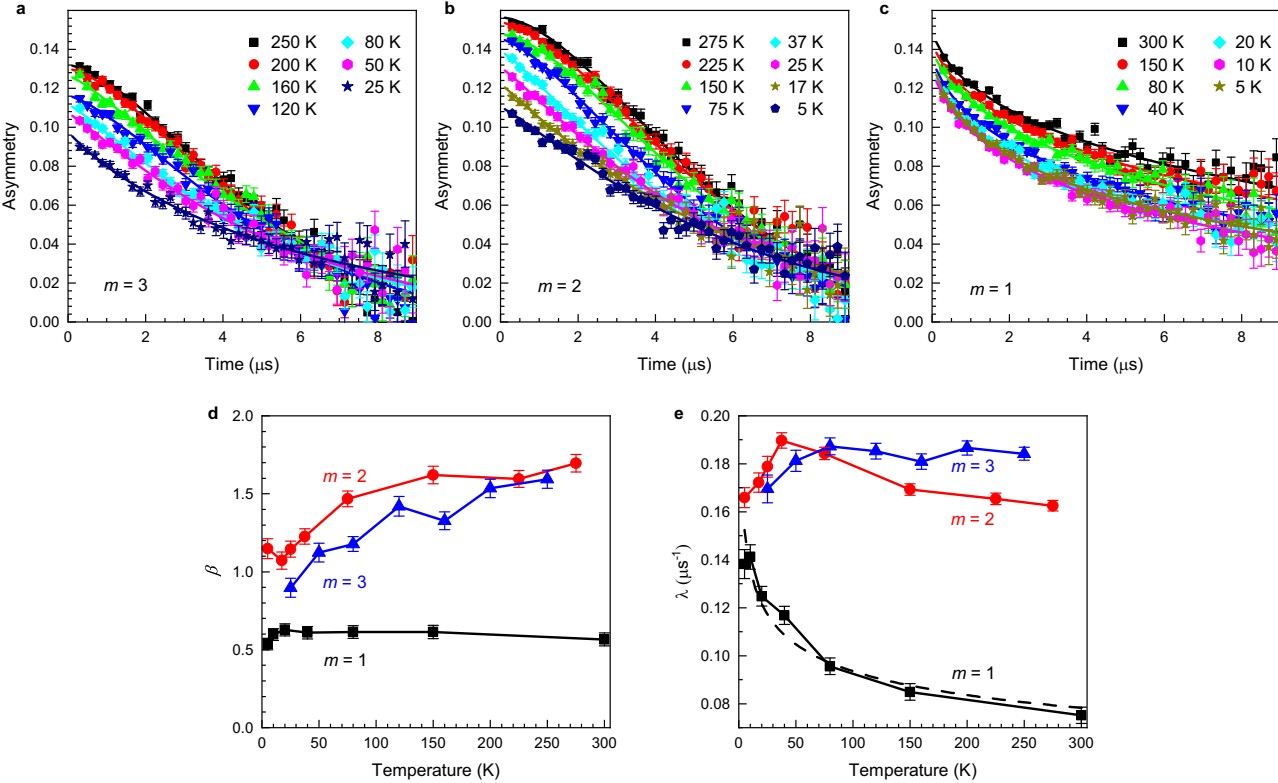

**Fig. 2 | Zero field muon spin rotation.** Time evolution of the zero-field muon spin polarization of $[(LaFeO_3)_m/(SrTiO_3)_5]_{10}$ superlattices with **a** $m = 3$, **b** $m = 2$, and **c** $m = 1$. Solid lines represent fit by the stretched exponential function $A_0 e^{-(\lambda t)^\beta}$. Exponent $\beta$ and depolarization rate $\lambda$ as a function of temperature are shown in panels **d** and **e**, respectively; solid lines are a guide to the eye. The dashed line in **e** represents fit to $m = 1$ data by $\lambda \sim T^{-\alpha}$. The error bars represent one standard deviation.

The obtained $\beta$ values of the thicker superlattices with $m = 2$ and $m = 3$ (see Fig. 2d), are at high temperatures >1.5 and close to the Gaussian profile ($\beta = 2$). Such a profile is usually associated with the damping on nuclear magnetic moments[30] typically visible in the paramagnetic phase, where the electronic moments are fluctuating so fast that the muons are effectively not sensitive to their presence. With decreasing temperature, values of $\beta$ decrease towards 1, and the initial asymmetry drops (see Supplementary Fig. 5), which is typical for an onset of a static magnetism[25,31]. In contrast, in the $m = 1$ superlattice, $\beta$ is in the whole temperature range close to 0.6, which indicates a qualitatively different magnetic state. Values of $\beta < 1$ were reported, e.g., for spin glass systems[32–34] and frustrated magnetic systems[26,35,36]. A similar qualitative difference between the $m = 1$ superlattice on the one side and the $m = 2$ and $m = 3$ superlattices on the other side can be seen in the values of $\lambda$ (see Fig. 2e), depicting that $\lambda$ for the $m = 1$ superlattice is two times smaller above 100 K compared to the $m = 2$ and $m = 3$ superlattices and significantly increases with decreasing temperature. In order to get more insight into the magnetic state of our superlattices, we have performed weak transverse and longitudinal field measurements discussed below.

**The magnetic volume fraction and the Néel temperature**
Muon spin rotation spectroscopy offers a way to determine the volume fraction of a magnetically ordered phase using a measurement where a weak external field is applied transverse to the muon spins. In a paramagnetic state, the fluctuation rate of electronic moments is too high to influence the muon spin direction, and thus, muons precess due to the external magnetic field, which is observed as an oscillation of the asymmetry. Figure 3a shows these oscillations in the weak transverse field asymmetry of the $m = 3$ superlattice at 300 K, which is at this temperature in the paramagnetic state. The solid line represents

a fit using the exponentially damped cosine function

$$A_{TF}(t) = A_0 \; e^{-\lambda_{TF} t} \cos[\gamma_\mu B_{ext} t + \phi], \tag{2}$$

where $A_0$ is the initial asymmetry, $\lambda_{TF}$ is the depolarization rate, $B_{ext}$ is the applied transverse field, and $\phi$ relates to the initial muon spin polarization. In an ordered magnetic phase, muon spins quickly depolarize because of the large static fields, which leads to the decrease of the oscillation amplitude, as can be seen in the asymmetry of the $m = 3$ superlattice at 10 K (see Fig. 3a). This reduction of the oscillation amplitude is a clear sign of the formation of a static magnetic order at low temperatures. The magnitude of this decrease yields the magnetic volume fraction, $f_{mag}$, which was calculated as

$$f_{mag}(T) = 1 - \frac{A_0(T)}{A_0(T_{high})}, \tag{3}$$

where $A_0(T_{high})$ is the mean of the initial weak transverse field asymmetry above 250 K in the expected paramagnetic state. We have determined $f_{mag}$ of our superlattices using measurements in a transverse field of 10 mT applied in a perpendicular direction to the superlattice surface. We corrected $f_{mag}$ for the muonium formation in $SrTiO_3$ (for details, see Supplementary Section 2.2).

The obtained $f_{mag}$ for the $m = 3$ superlattice (see Fig. 3b) exhibits an onset near 175 K and increases with the lowering of the temperature, which is typical for a magnetically ordered state. At 10 K, $f_{mag}$ is above 0.6, which is more than the LaFeO$_3$ volume fraction, $f_{V,m=3} = 3/8$, which depicts that the antiferromagnetic state is well developed with some stray fields reaching into SrTiO$_3$ layers. The stray fields are likely caused by the small canting of LaFeO$_3$ moments[15]. In the $m = 2$ superlattice, $f_{mag}(T)$ exhibits a weak increase below 200 K, a

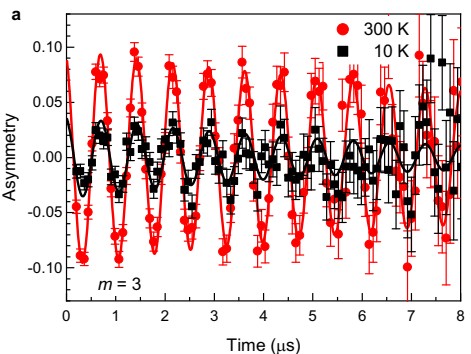
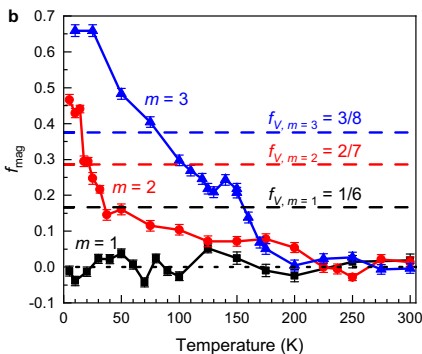
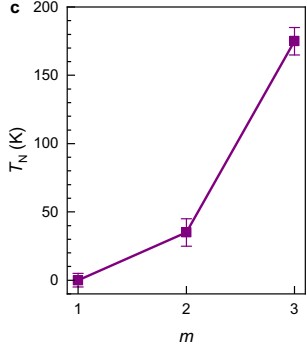

**Fig. 3 | The magnetic volume fraction and the Néel temperature. a** Time evolution of the muon spin polarization in the weak transverse field of 10 mT in the $[(LaFeO_3)_m/(SrTiO_3)_5]_{10}$ superlattice with $m = 3$ at 300 and 10 K shown with a fit (solid lines) using Eq. (2). **b** Magnetic volume fraction, $f_{mag}$, of the superlattices evaluated from the measurement in the weak transverse field. Horizontal dashed lines represent the volume fraction of $LaFeO_3$, $f_V$. **c** The Néel temperature with respect to $m$ determined from panel (**b**). The error bars in panels (**a**) and (**b**) represent one standard deviation, and the error bars in panel (**c**) were estimated from data in panel (**b**).

sharp onset below 35 K and reaches above 0.4 at 5 K. This value is again larger than $LaFeO_3$ volume fraction $f_{V,m=2} = 2/7$, demonstrating that even in this superlattice with only two monolayers of $LaFeO_3$, the antiferromagnetic state is well developed at 5 K, although with significantly reduced $T_N$ to 35 K. In contrast, $f_{mag}$ of the $m = 1$ superlattice is zero within the experimental error bars down to the lowest measured temperature of 5 K, showing the absence of formation of a static order in the measured temperature range. The qualitative difference between $f_{mag}$ of $m = 3$ and $m = 2$ superlattices on the one hand and of the $m = 1$ superlattice on the other hand again depicts the qualitative difference in their magnetic ground state.

Since muons stop in the superlattices at various sites, it is not possible to determine from the muon spin rotation data whether the order is ferromagnetic or antiferromagnetic. Because bulk $LaFeO_3$ is a G-type antiferromagnet, it is reasonable to expect that if the order in the superlattices was antiferromagnetic, its transition temperature would monotonically increase with increasing $m$ as the properties approach those of bulk $LaFeO_3$. Since the observed dependence of the transition temperature with increasing $m$ is indeed monotonic and rapidly increasing (see Fig. 3c), this indicates that the observed order in the $m = 2$ and $m = 3$ superlattices is antiferromagnetic; however, the data do not exclude other static magnetic orders. In our superlattices with $m \leq 3$, $T_N$ is still much smaller compared to the bulk value of 740 K. To some extent, this reduction can be due to a change of valency of Fe due to proximity to Sr ions at the interface between $LaFeO_3$ and $SrTiO_3$. This effect is the strongest in the $m = 1$ superlattice where the iron oxide layer is formed only by one LaO and one $FeO_2$ layer (see Fig. 1a), and thus Fe ions are surrounded equally by La and Sr ions. Nevertheless, since bulk $La_{0.5}Sr_{0.5}FeO_3$ is still antiferromagnetic with $T_N$ of about 250 K[37], we conclude that the strong reduction of $T_N$ of $m = 2$ and $m = 1$ superlattices is predominantly due to the dimensional crossover rather than due to the change of the Fe valency.

**Differentiation between static and dynamic magnetism**
The zero field and the weak transverse field data indicate that there is no magnetic order in the $m = 1$ superlattice down to 5 K. This could be explained by two scenarios: a static disorder (e.g., due to structural defects as ionic diffusion) or dynamic fluctuations of the electronic moments. Muon spin rotation spectroscopy offers a way to unequivocally differentiate between static magnetism and dynamically fluctuating fields by measurements in the magnetic field longitudinal to the muon spin direction. In the presence of static magnetism, muons in the longitudinal field with a magnitude much larger than that of the local fields essentially do not precess (so-called decouple from the local fields) and thus do not depolarize in contrast to the zero field

measurements. However, if the local fields are fluctuating, they cause a random muon spin-flip (a transition between the Zeeman split energy levels) and give rise to the muon-spin depolarization even in the longitudinal field, essentially the same as in zero field[24]. Time evolutions of muon spin polarization in the $m = 1$ superlattice at 5 K in several longitudinal fields are shown in Fig. 4a; data are normalized as detailed in Supplementary Section 2.3. The asymmetry increases between zero field and 2.5 mT, which is caused by the decoupling of the muon spins from the static nuclear moments of $SrTiO_3$[30]. However, for higher fields between 2.5 and 125 mT, the asymmetry is essentially field-independent and exhibits at 8 μs considerable depolarization to about 40% of the initial value. Such a significant depolarization independent of the longitudinal field is a hallmark of fluctuating electronic moments (see, e.g., ref. 36).

We have modeled the normalized asymmetry in the longitudinal field, $A_{LF}^N$, as a sum of the theoretical Gaussian Kubo–Toyabe functions for dynamic fluctuations, $P_{dyn}$[38], and for the static disorder, $P_{stat}$[30]

$$A_{LF}^N = cP_{dyn} + (1 - c)P_{stat},  \quad (4)$$

where $c$ is the fraction of the fluctuating part (for details, see Supplementary Section 2.3). The global fit for all longitudinal fields $B_{ext}$, see solid lines in Fig. 4a, yields the fraction $c = 0.64 \pm 0.06$ and the distribution of the static disordered moments $\sigma_s/\gamma_\mu = 0.32 \pm 0.08$ mT. The functions $P_{dyn}$ displayed in Fig. 4b for the obtained parameter values are essentially field-independent and vanish at 8 μs. In contrast, $P_{stat}$, displayed in Fig. 4c, sensitively depends on the external magnetic field. This difference allows the model to discern between statically disordered and dynamically fluctuating moments. The obtained value of $\sigma_s/\gamma_\mu = 0.32 \pm 0.08$ mT is typical for nuclear moments[30]. The fact that we can fit the data with the model yielding such a small value of $\sigma_s/\gamma_\mu$ at all external fields is incompatible with the picture of statically disordered iron moments with local fields expected to be in the order of 100–250 mT[39]. If iron moments were static, the increase of the longitudinal field between 10 and 125 mT would lead to a significant increase in asymmetry[24]. The field-independent asymmetry exhibiting such a considerable depolarization for fields above 2.5 mT can be explained only as a consequence of the fluctuating iron moments. The suppression of the static magnetic order in the $m = 1$ superlattice due to structural defects would most likely lead to statically disordered magnetic moments at low temperatures. The observation that the iron magnetic moments fluctuate at 5 K indicates that this scenario is highly unlikely.

It is interesting to review the zero-field data in the context of the fluctuating scenario of the $m = 1$ superlattice. The convex profile of the

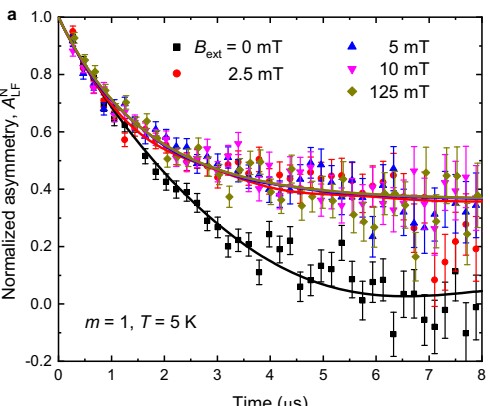
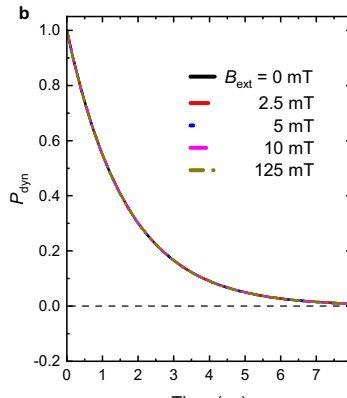
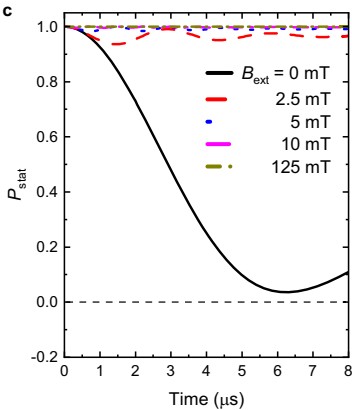

**Fig. 4 | Differentiation between the static and dynamic magnetism. a** Time evolution of normalized muon spin polarization, $A_{LF}^{N}$, of the $m = 1$ superlattice at 5 K for various applied longitudinal magnetic fields. Error bars represent one standard deviation. The solid lines represent fit using a model given by Eq. (4). The significant decrease of asymmetry at high fields is a hallmark of dynamic magnetism. Panels **b** and **c** display the theoretical Gaussian Kubo-Toyabe functions used in the fit for dynamically fluctuating moments, $P_{dyn}$, and for static disordered moments, $P_{stat}$, respectively.

zero field data ($\beta \approx 0.6$) of the $m = 1$ superlattice (see Fig. 2c), indicates that the muons are significantly depolarized by the electronic moments. This is in contrast to the depolarizations of the $m = 2$ and $m = 3$ superlattices (see Fig. 2a and b), respectively, that exhibit concave shape above $T_N$. The concave (Gaussian) shape of depolarization is typically interpreted as due to the depolarization on the much weaker nuclear moments because, in the paramagnetic phase, the electronic moments fluctuate too fast to be followed by muons[30]. This, however, indicates that the electronic moments in the $m = 1$ superlattice are fluctuating significantly slower compared to the $m = 2$ and $m = 3$ superlattices at high temperatures. This can be understood since the Mermin–Wagner theorem states that the antiferromagnetic order in 2D is destroyed by the long-wavelength (therefore slow) fluctuations, which in the 3D case have a much smaller magnitude. Therefore, the observation of $\beta \approx 0.6$ is indicative of strong in-plane magnetic correlations that persist to very high temperatures in the range of 300 K. This interpretation is, in addition, supported by the observed increase of the depolarization rate $\lambda$ of the $m = 1$ superlattice with decreasing temperature (see Fig. 2e). This increase can be explained by slowing down of fluctuations as the magnetic system is approaching the ordered state (see, e.g., ref. 40). Mermin–Wagner theorem proposes that in the case of a 2D magnetic system, the ordered state occurs at zero temperature[8]. Indeed, the temperature dependence $\lambda$ for the $m = 1$ superlattice can be reasonably well modeled with a power law $\lambda \sim T^{-\alpha}$[31], where $\alpha = 0.16 \pm 0.02$ (see dashed line in Fig. 2e).

In summary, the muon spin rotation data in zero, transverse, and longitudinal fields consistently show that (i) $m = 3$ and $m = 2$ superlattices exhibit a long-range antiferromagnetic order with $T_N$ of 175 and 35 K, respectively, (ii) that the magnetic properties of the $m = 1$ superlattice are qualitatively different with no long-range order down to the lowest measured temperature of 5 K and (iii) that at this temperature, the electronic moments are fluctuating rather than statically disordered. These findings point towards a dimensional magnetic crossover where for the superlattice with a single monolayer of iron oxide, the static antiferromagnetic order is lost due to enhanced magnitude of long-wavelength spin fluctuations, as expected from the Mermin–Wagner theorem. Note, however, that our results need not be in stark disagreement with the work of Jenkins et al.[11] predicting a stabilization of the magnetic order in 2D finite-size lab samples because (i) their calculations were performed for systems with four orders of magnitude smaller size than our samples and (ii) there is always a possibility that there is a static order in our $m = 1$ superlattice below 5 K, currently the lowest

achievable temperature in the low-energy muon spin rotation instrument[22].

## Methods

### Sample growth and characterization
Superlattices were fabricated by pulsed laser deposition on $10 \times 10 \text{ mm}^2$ TiO-terminated $SrTiO_3$ (001) substrates. The deposition temperature of the substrates was 570 °C, and the background oxygen pressure was 0.01 mbar. The thickness of layers was in situ controlled by reflection of high-energy electron diffraction. The samples were annealed ex-situ in an oxygen atmosphere at 550 °C to reduce the concentration of oxygen vacancies. We fabricated sets of 3–4 samples of each superlattice that formed a sample mosaic to improve the signal-to-noise ratio of the muon spin rotation data. The structural quality of the superlattices was characterized using an atomic force microscope (Bruker Dimension Icon) and an X-ray diffractometer (Rigaku Smartlab). Atomic force microscope images were analyzed by Gwyddion software[41] and the superlattice structure shown in Fig. 1a was created using VESTA software[42].

### Low-energy muon spin rotation
Low-energy muon spin rotation experiments were performed at the $\mu$E4 beamline of the Swiss Muon Source at Paul Scherrer Institute, Villigen. We used 2 keV muon beam that results in an implantation profile, where most of the muons stop in the superlattices (see Supplementary Fig. 4). $\mu$SR data were analyzed using musrfit[43].

## Data availability
All relevant data are available from the authors. Alternatively, the $\mu$SR data generated in this study have been deposited in the PSI Public Data Repository[44].

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

## Acknowledgements

We thank C. Bernhard, J. Chaloupka, G. Jackeli, J. Kuneš, D. Munzar, and K. Penc for fruitful discussions. We thank K. Bernatova (Tescan Orsay Holding) and S. Dinara for their help with sample preparation. A.D. and M.K. acknowledge the financial support by the project Quantum Materials for Applications in Sustainable Technologies, CZ.02.01.01/00/22_008/0004572, by the Czech Science Foundation (GACR) under Project No. GA20-10377S and CzechNanoLab project LM2023051 funded by MEYS CR for the financial support of the measurements/sample fabrication at CEITEC Nano Research Infrastructure. B.Ch. acknowledge the financial support by the National Natural Science Foundation of China (Grant No. 12104157) and Shanghai Sailing Program (Grant No. 21YF1410700). Part of this work is based on experiments performed at the Swiss Muon Source S$\mu$S, Paul Scherrer Institute, Villigen, Switzerland.

## Author contributions

A.D and M.K. conceived and designed the study; M.K. developed the samples, and performed characterization measurements; M.K., A.D., A.S., Z.S., and T.P. performed muon measurements, in which A.S., Z.S., and T.P. provided beamline support; B.C. and G.K. helped designing samples; A.D. and M.K. wrote the manuscript. All authors contributed to the discussions and to the final manuscript.

## Competing interests

The authors declare no competing interests.
