## [Peer Review File · Nature Communications]

Reviewers' Comments:

Reviewer #1:

Remarks to the Author:

This manuscript reports on the investigation of magnetic order in low-dimensional samples using low-energy muon spin rotation. The samples are $(\text{LaFeO}_3)_m/(\text{SrTiO}_3)_5$ multilayers in which the magnetic films are 1, 2 or 3 monolayers thick. The main finding reported there is the existence of a dimensional crossover for antiferromagnetic ordering, already at the thickness of 2 monolayers. The authors find finite transition temperatures for $m=2$ and $m=3$, whereas the $m=1$ samples do not show evidence of magnetic ordering even for the lowest temperature probed (5 K). They also show evidence that the magnetic state of the $m=1$ samples is one in which the Fe magnetic moments fluctuate dynamically, in contrast with the alternative picture of static disorder. They assert that their findings are compatible with the predictions of the Mermin-Wagner theorem.

The text is well-written, the exposition of the experimental technique and of the physical interpretations of the results are detailed and clear. The results reported are very relevant to the broad field of low-dimensional magnetism, in particular to the discussion of the applicability of the MW theorem to real, finite size samples. Regarding the latter point, I would like to remark that the lateral sizes of the samples in this manuscript are 4 orders of magnitude larger than the largest systems considered in ref. 11. I believe this still leaves some room for the possibility that, in smaller samples or at lower temperatures than 5 K, deviations from MW behavior is still observed. I would like the authors to consider including this discussion in their text.

There is an issue that is confusing me: on lines 170-174, the authors describe the results of μSR in zero magnetic field. They state that, for $m=1$, the results indicate slow fluctuations of the Fe moments, even at high temperatures. Naively, I would expect slow fluctuations to be associated with long-range correlations, which would be incompatible with the fact that the $m=1$ samples do not show long-range order down to 5 K. Could the authors please clarify?

Regarding the nature of the magnetic state of the $m=1$ samples, I wonder if more information could be extracted from the present results. For instance, could the relaxation rate as a function of temperature reveal some kind of transition in the nature of the spin fluctuations (from thermal to quantum, for example)?

In summary, I believe the manuscript deserves to be published in Nature Communications once the issues raised above are considered.

Reviewer #2:

Remarks to the Author:

In this paper, the dimensionality dependence of magnetism of LaFeO_3 . The authors grow superlattices of this materials combined with SrTiO_3 and tune the dimensionality of LFO by choosing single, bi- or trilayers of this materials (separated each time with 5 layers of STO – 10 repetition in total are made in order to ensure sufficient volume to obtain measurable signals). The magnetic order of the samples is investigated with low energy muon spin rotation and the main finding is a suppression of static antiferromagnetic ordering in the single layer limit of LFO. This is interpreted as a realization of the Mermin-Wagner theorem which predicts a suppression of long-range magnetic order due to enhanced magnetic fluctuations in reduced dimensions.

This is without any doubt a timely paper as the recent years have seen a surge of work on low dimensional magnetic materials (in particular in the context of van der Waals materials). Addressing the case of antiferromagnets (which do not have a net magnetic moment) is particularly challenging given that most relevant experimental methods – starting with inelastic neutron scattering – require large sample volume which are not compatible with 2D limit. In this

respect the choice of low energy μ SR is a clever one, and provides convincing results. What is less convincing in the paper, and which prevents me to recommend it for publication in Nature communications, is that the evidence that the disappearance of the AF order is solely driven by the reduce dimensionality is rather scarce.

The main 'bulk' reference for the magnetism in LFO (ref. 14) is based on neutron powder diffraction, and quite clearly the behavior of the trilayer departs already significantly from this. The Néel temperature is for instance strongly reduced (from 740K in the bulk to less than 200K in the trilayer). This illustrates the main issue I have with this work, which is, to which extent to the difference between the film LFO and the bulk ones. Is the reduction of TN for instance related to strain? To interfacial charge transfer? To different oxygen contents?

There is a lot of missing characterization of the samples (to give a few examples: reciprocal space mapping to assess the strain state, x-ray absorption to cross-check the valence state of Fe etc...) which should be included to ensure that the disappearance of the long-range magnetic order is indeed only driven by the reduced dimensionality. I would also strongly recommend to include a 'bulk' reference, that is data from a thin film of LaFeO₃ only.

Without these, while I acknowledge that the results are intriguing, I cannot recommend their publication.

Reviewer #3:

Remarks to the Author:

Spin fluctuation at low dimension is crucial in stabilizing magnetic ordering and controlling of the moment. Michal Kiaba and colleagues are able to utilize heterostructure engineering to suppress the AFM ordering by lowering the dimensionality of their LFO/STO thin films. As the superlattice approaches 2D limit, the authors are able to observe the absence of long-range magnetic ordering by muon spin rotation spectroscopy. More importantly, the authors are able to identify that the superlattice is fluctuation dominated which is very challenging in the study of Mermin-Wagner behavior. While the approach by the authors looks utmost like Nat. Phys 14, 806–810 (2018), the fact the authors can present a 2D system without long-range order is very appreciable. Since anisotropy in most materials is large enough to suppress the fluctuation leading to a transition like XY or Ising model, the $m=1$ superlattice shown by the author is surprisingly isotropic, at least the fluctuation is much stronger than anisotropy. Unfortunately, the authors do not seem to have any data or discussion regarding this, particularly since the authors have mentioned the canting of moment several times. On the other hand, the $m=3$ and $m=2$ data seems to contribute very little to the whole manuscript. The authors mentioned that the magnetic state changes from $m=3$ to 2 but there is no data or discussion about it. I also fail to understand the fluctuation that the authors are trying to refer to is thermal driven or quantum. Overall, to be able to create a 2D system without long range order and prove it is primally fluctuation driven is truly amazing work. However, the current manuscript appears to be a publication for Scientific Reports instead of Nature Communication. I would consider recommendation with a major revision.

I also list some of my concerns regarding some details in the manuscript.

Major

1. While the XRD pattern in Fig.1 C shows some promising features like well-defined film peak next to the substrate peak, very clear fringes and some superlattice peaks, there are some details that concern me.

The authors can show SL_n, but there is some inconsistency for (SL-2, SL2) among $m=1,2,3$. The SL2 peak can be clearly shown in $m=1,2,3$ and it is closer to SL0 for larger m which is expected. But the quality of SL-2 is much worse, and the position of SL-2 seems to move away from SL0 as m increases which does not make sense.

I noticed that in the supplementary the authors use $a_{pc}=4.009$ angstrom for LFO epitaxial film but in some other work like [J.Phys.:Condens.Matter20(2008)264014(10pp) ;Appl. Phys. Lett. 102, 081904 (2013)] it is shown that LFO on STO(001) has a c-axis lattice constant of 3.95 angstrom.

Even with the bulk value given from the authors' previous work, a fully strained LFO on STO(001) should have a c-axis lattice constant ~ 3.98 angstrom.

2. The zero-field muon spin rotation creates a vast confusion for me. The first argument of electron spin fluctuation is very slow at high temperature does not make much sense to me. In addition, Fig. 2(d) seems to categorize $m=1$ case as a spin glass, which I do not think is consistent with what the authors are trying to show throughout the manuscript.

3. From Line 294, I fail to see the logic between the assumption that the sample is antiferromagnetic because the ordering temperature increases with m . I do not quite understand how this observation excludes ferromagnetism. I agree with the conclusion about antiferromagnetism and reduction of dimensionality but the way that the authors presented is a bit confusing.

4. For static and dynamic analysis, I agree with the authors that the longitudinal field measurement could be used to distinguish short-range order and dynamic moment, but the data the author presented is very confusing. The data clearly show that at least there is some static magnetic moment which is also the authors' argument. However, I fail to understand why the static moment must be from the nuclear moment of SrTiO₃. Is it from Sr, Ti? Can it come from La? Why cannot be from electron spin moment of hybridized O? I would suggest the authors to present the argument more clearly and accessible for general reader since the comparison of static and dynamic is very important in this manuscript.

Minor

1. Typo on line 085 should be 'rare' instead of 'rear'

Dear referees,

thank you for your very valuable remarks. We have considered them seriously and, in most cases, changed the manuscript correspondingly, which, in our view, led to improvements of the manuscript. The responses to the raised points are displayed in red following each remark.

REVIEWER COMMENTS

Reviewer #1 (Remarks to the Author):

This manuscript reports on the investigation of magnetic order in low-dimensional samples using low-energy muon spin rotation. The samples are $(\text{LaFeO}_3)_m/(\text{SrTiO}_3)_5$ multilayers in which the magnetic films are 1, 2 or 3 monolayers thick. The main finding reported there is the existence of a dimensional crossover for antiferromagnetic ordering, already at the thickness of 2 monolayers. The authors find finite transition temperatures for $m=2$ and $m=3$, whereas the $m=1$ samples do not show evidence of magnetic ordering even for the lowest temperature probed (5 K). They also show evidence that the magnetic state of the $m=1$ samples is one in which the Fe magnetic moments fluctuate dynamically, in contrast with the alternative picture of static disorder. They assert that their findings are compatible with the predictions of the Mermin-Wagner theorem.

The text is well-written, the exposition of the experimental technique and of the physical interpretations of the results are detailed and clear. The results reported are very relevant to the broad field of low-dimensional magnetism, in particular to the discussion of the applicability of the MW theorem to real, finite size samples.

1. Regarding the latter point, I would like to remark that the lateral sizes of the samples in this manuscript are 4 orders of magnitude larger than the largest systems considered in ref. 11. I believe this still leaves some room for the possibility that, in smaller samples or at lower temperatures than 5 K, deviations from MW behavior is still observed. I would like the authors to consider including this discussion in their text.

Response:

We agree with the reviewer that there is still room for possible deviation from Mermin-Wagner behavior at lower temperatures and that the calculations from ref. 11 were made for a significantly smaller sample size. We have therefore added a note about this issue to the manuscript in the conclusion at the end of the main part.

2. There is an issue that is confusing me: on lines 170-174, the authors describe the results of μSR in zero magnetic field. They state that, for $m=1$, the results indicate slow fluctuations of the Fe moments, even at high temperatures. Naively, I would expect slow fluctuations to be associated with long-range correlations, which would be incompatible with the fact that the $m=1$ samples do not show long-range order down to 5 K. Could the authors please clarify?

Response:

We agree with the referee that this conclusion might look confusing. We have, therefore, added a paragraph at the end of the manuscript, right before the summary, that discusses this point in more detail and removed the short conclusion located originally on lines 170-174. We still think that the convex shape of the $m = 1$ superlattice is indeed indicative of a relatively smaller fluctuation rate in $m = 1$ superlattice compared to the paramagnetic phase of the $m = 2$ and $m = 3$. We agree with the referee that slow fluctuations are associated with long-range AF correlations. However, such correlations can indeed occur in $m = 1$ superlattices because the Mermin-Wagner theorem predicts that the AF order in 2D should occur at absolute zero. As the temperature decreases, the fluctuation rate should decrease, which is indeed observed since the depolarization rate increases as T decreases. Note, however, that despite being very interesting, this is not the key argument in our paper. Our main point is based on the weak transverse field and longitudinal field data.

3. Regarding the nature of the magnetic state of the $m=1$ samples, I wonder if more information could be extracted from the present results. For instance, could the relaxation rate as a function

of temperature reveal some kind of transition in the nature of the spin fluctuations (from thermal to quantum, for example)?

Response:

Inspired by this comment, we have modeled the relaxation rate for the $m = 1$ samples with a power law ($\lambda \sim T^{-\alpha}$) and discussed it in the added paragraph at the end of the manuscript. We have moved the graph of λ with the model to the main part of the paper to the Fig. 2(e). However, we do not think that with this relatively sparse temperature dataset, we can get more insight into the nature of the fluctuations (e.g. thermal vs quantum). This topic is indeed very interesting however would require more experimental work.

In summary, I believe the manuscript deserves to be published in Nature Communications once the issues raised above are considered.

Reviewer #2 (Remarks to the Author):

In this paper, the dimensionality dependence of magnetism of LaFeO₃. The authors grow superlattices of this materials combined with SrTiO₃ and tune the dimensionality of LFO by choosing single, bi- or trilayers of this materials (separated each time with 5 layers of STO – 10 repetition in total are made in order to ensure sufficient volume to obtain measurable signals). The magnetic order of the samples is investigated with low energy muon spin rotation and the main finding is a suppression of static antiferromagnetic ordering in the single layer limit of LFO. This is interpreted as a realization of the Mermin-Wagner theorem which predicts a suppression of long-range magnetic order due to enhanced magnetic fluctuations in reduced dimensions.

This is without any doubt a timely paper as the recent years have seen a surge of work on low dimensional magnetic materials (in particular in the context of van der Waals materials). Addressing the case of antiferromagnets (which do not have a net magnetic moment) is particularly challenging given that most relevant experimental methods – starting with inelastic neutron scattering - require large sample volume which are not compatible with 2D limit. In this respect the choice of low energy muSR is a clever one, and provides convincing results.

1. What is less convincing in the paper, and which prevents me to recommend it for publication in Nature communications, is that the evidence that the disappearance of the AF order is solely driven by the reduce dimensionality is rather scarce.

The main ‘bulk’ reference for the magnetism in LFO (ref. 14) is based on neutron powder diffraction, and quite clearly the behavior of the trilayer departs already significantly from this. The Néel temperature is for instance strongly reduced (from 740K in the bulk to less than 200K in the trilayer). This illustrates the main issue I have with this work, which is, to which extent to the difference between the film LFO and the bulk ones. Is the reduction of TN for instance related to strain? To interfacial charge transfer? To different oxygen contents? There is a lot of missing characterization of the samples (to give a few examples: reciprocal space mapping to assess the strain state, x-ray absorption to cross-check the valence state of Fe etc. . .) which should be included to ensure that the disappearance of the long-range magnetic order is indeed only driven by the reduced dimensionality. I would also strongly recommend to include a ‘bulk’ reference, that is data from a thin film of LaFeO₃ only. Without these, while I acknowledge that the results are intriguing, I cannot recommend their publication.

Response:

We thank the referee for acknowledging that our paper is timely. The referee in addition claims that the evidence that the disappearance of the AF order is driven solely by the reduced dimensionality is rather scarce and proposes to compare thin films of LFO and bulk samples. In addition, the referee proposes to do more characterization e.g. reciprocal space mapping etc.

However, we disagree with this critique because of the following reasons. First, any comparison between thin film and bulk LFO will not give the answer to the question concerning the dimensionality crossover, which happens in our superlattices between $m = 1$ and $m = 2$. Secondly, the referee proposes to do additional characterization to check the influence of strain, interfacial charge transfer or oxygen content. However, these characterizations, although potentially

valuable for a larger picture, are not probing the magnetic state of the superlattices, which is the topic of this paper and of our conclusion. On the contrary, our work presents results of a very sensitive magnetic probe (muon spin rotation) showing that the superlattices with $m = 3$ and $m = 2$ superlattices exhibit static magnetism in contrast to the $m = 1$ superlattice where the magnetic moments are fluctuating down to the lowest achievable temperature. We do not see any more direct way of probing the influence of dimensionality on the magnetic states of the superlattices than what we present in our study. The only issue here might be the interfacial doping due to the proximity of the Sr ions, which is the strongest in the $m = 1$ superlattice. However, this issue was already discussed in the original version of our manuscript at the end of the discussion of the weak transverse field data, where we noted that:

“since bulk $\text{La}_{0.5}\text{Sr}_{0.5}\text{FeO}_3$ is still antiferromagnetic with T_N of about 250 K [Matsuno1999], we conclude that the strong reduction of T_N of $m = 2$ and $m = 1$ superlattices is predominantly due to the dimensional crossover rather than due to the change of the Fe valency.”

The referee additionally raises concern about the reduced T_N of our superlattice with $m = 3$ compared to the bulk value of 740 K. However, the opposite is actually the case: despite having only three monolayers of LFO, the superlattice with $m = 3$ has already T_N of 175 K. Note that the critical temperature of magnetic thin films is typically reduced from the bulk values as thickness decreases; see the references 1-7 in our paper. The fact that the superlattice with $m = 3$ has T_N already about 175 K is actually an indication of the very high quality of our superlattices.

Nevertheless, in order to present more characterization and to clarify to the scientific community the quality of our samples, we have added to the supplementary online material symmetric and asymmetric X-ray maps demonstrating that our superlattices are fully strained, and in particular, that there is no difference in degree of strain between the $m = 1$ and $m = 2$ superlattice.

Reviewer #3 (Remarks to the Author):

Spin fluctuation at low dimension is crucial in stabilizing magnetic ordering and controlling of the moment. Michal Kiaba and colleagues are able to utilize heterostructure engineering to suppress the AFM ordering by lowering the dimensionality of their LFO/STO thin films. As the superlattice approaches 2D limit, the authors are able to observe the absence of long-range magnetic ordering by muon spin rotation spectroscopy. More importantly, the authors are able to identify that the superlattice is fluctuation dominated which is very challenging in the study of Mermin-Wagner behavior. While the approach by the authors looks utmost like Nat. Phys 14, 806–810 (2018), the fact the authors can present a 2D system without long-range order is very appreciable. Since anisotropy in most materials is large enough to suppress the fluctuation leading to a transition like XY or Ising model, the $m=1$ superlattice shown by the author is surprisingly isotropic, at least the fluctuation is much stronger than anisotropy. Unfortunately, the authors do not seem to have any data or discussion regarding this, particularly since the authors have mentioned the canting of moment several times. On the other hand, the $m=3$ and $m=2$ data seems to contribute very little to the whole manuscript. The authors mentioned that the magnetic state changes from $m=3$ to 2 but there is no data or discussion about it. I also fail to understand the fluctuation that the authors are trying to refer to is thermal driven or quantum.

Overall, to be able to create a 2D system without long range order and prove it is primarily fluctuation driven is truly amazing work. However, the current manuscript appears to be a publication for Scientific Reports instead of Nature Communication. I would consider recommendation with a major revision.

Response:

We thank the referee for recognizing that the demonstration of the disappearance of the long-range order in 2D is appreciable. Concerning the issue of the anisotropy, we cite at the beginning of the manuscript the results of neutron diffraction, see Ref. [14] demonstrating that the spin Hamiltonian is within the errorbars completely isotropic.

We do not understand why the referee states that the data of $m=3$ and $m=2$ contribute very little to the whole manuscript since we report about their zero field (see Fig. 2) and weak transverse field measurements (see Fig. 3), demonstrating that their magnetism is static. This is an important finding in our work since we show that, in contrast, in $m = 1$ superlattice, the static magnetism is absent (see

Fig. 3) and that the magnetic moments fluctuate down to the lowest measurable temperature (see Fig. 4.)

The issue of whether the fluctuations are quantum or thermal is potentially very interesting but very complex. Note that even in 3D, the antiferromagnetic state intrinsically exhibits quantum fluctuations since its ground state is a superposition of the two allowed spin states (up, down, up, down), and (down, up, down, up). And at any finite temperature, the system will exhibit a degree of thermal fluctuations as well. Therefore, in our samples, there is presumably a combination of thermal and quantum fluctuations. The increase of the relaxation rate with decreasing temperature of the $m = 1$ superlattice (see Fig. 2(e) in the updated manuscript) is, in our view, indicative of slowing down of spin fluctuations with temperature - and, therefore, is due to the thermal fluctuations. However, this does not exclude that there is (and there should be as discussed above) a degree of quantum fluctuations that should be, probably (?) temperature independent. We think that the topic of disentangling the quantum and thermal fluctuations is very interesting but goes beyond the scope of the present manuscript. After establishing the present findings, we may come back to this topic in future research.

I also list some of my concerns regarding some details in the manuscript. Major

1. While the XRD pattern in Fig.1 C shows some promising features like well-defined film peak next to the substrate peak, very clear fringes and some superlattice peaks, there are some details that concern me. The authors can show SL_n, but there is some inconsistency for (SL₋₂, SL₂) among $m=1,2,3$. The SL₂ peak can be clearly shown in $m=1,2,3$ and it is closer to SL₀ for larger m which is expected. But the quality of SL₋₂ is much worse, and the position of SL₋₂ seems to move away from SL₀ as m increases which does not make sense. I noticed that in the supplementary the authors use $a_{pc}=4.009$ angstrom for LFO epitaxial film but in some other work like [J.Phys.:Condens.Matter20(2008)264014(10pp) ;Appl. Phys. Lett. 102, 081904 (2013)] it is shown that LFO on STO(001) has a c-axis lattice constant of 3.95 angstroms. Even with the bulk value given from the authors' previous work, a fully strained LFO on STO(001) should have a c-axis lattice constant 3.98 angstrom.

Response:

We thank the referee for pointing this out. Indeed in Fig.1(c) there was an inconsistency for (SL₋₂,SL₂) among the samples. The arrow that was pointing in the direction of SL₋₂ was not pointing correctly. The peaks on which the arrow was pointing were not SL₋₂ peaks but peaks probably due to multiple diffractions in superlattices that occasionally occur at low X-ray intensities. To not confuse readers we decided to show data only from SL₋₁ to SL₁ peaks.

As for the out-of-plane parameter of LaFeO₃, we remeasured X-ray diffraction on an LFO thin film annealed in the same way as the superlattices and obtained the out-of-plane pseudocubic lattice parameter $a_{pc}=4.002$ Å. The value $a_{pc}=4.009$ Å reported in the first draft corresponded to an unannealed LFO film. A higher lattice constant could point to remaining oxygen vacancies in the film, but still, some works show lattice constants higher, e.g. 4.03 Å or 4.04 Å see Appl. Phys. Lett. 113, 072901(2018) and Nano Lett. 2010, 10, 11, 4578–4583. We also measured the reciprocal space map to ensure that the film is pseudomorphic. Both X-ray diffraction and reciprocal space maps are now shown in Supplementary Fig.1(b) and Supplementary Fig. 2, respectively. In the calculation of bilayer thickness, we now use the LaFeO₃ lattice of 4.002 Å. Note however, that the difference between $a_{pc}=4.009$ Å and $a_{pc}=4.002$ Å is very small (on the level of 0.1%) and does not noticeably change the comparison with the experimental values shown in Supplementary Fig.1(a).

2. The zero-field muon spin rotation creates a vast confusion for me. The first argument of electron spin fluctuation is very slow at high temperature does not make much sense to me. In addition, Fig. 2(d) seems to categorize $m=1$ case as a spin glass, which I do not think is consistent with what the authors are trying to show throughout the manuscript.

Response:

We repeat the answer to a similar question from Referee #1.

“We agree with the referee that this conclusion might look confusing. We have, therefore, added

a paragraph at the end of the manuscript, right before the summary, that discusses this point in more detail and removed the short conclusion located originally on lines 170-174. We still think that the convex shape of the $m = 1$ superlattice is indeed indicative of a relatively smaller fluctuation rate in $m = 1$ superlattice compared to the paramagnetic phase of the $m = 2$ and $m = 3$. We agree with the referee that slow fluctuations are associated with long-range AF correlations. However, such correlations can indeed occur in $m = 1$ superlattices because the Mermin-Wagner theorem predicts that the AF order in 2D should occur at absolute zero. As the temperature decreases, the fluctuation rate should decrease, which is indeed observed since the depolarization rate increases as T decreases. Note, however, that despite being very interesting, this is not the key argument in our paper. Our main point is based on the weak transverse field and longitudinal field data.”

Additionally, we have removed from Fig. 2(d) the bars corresponding to diverse magnetic states, including spin glass, and rather discuss the beta values reported in the literature in the text. Indeed, we do not want to categorize our sample as spin glass.

3. From Line 294, I fail to see the logic between the assumption that the sample is antiferromagnetic because the ordering temperature increases with m . I do not quite understand how this observation excludes ferromagnetism. I agree with the conclusion about antiferromagnetism and reduction of dimensionality but the way that the authors presented is a bit confusing.

Response:

The idea behind the argument is following: since LFO is a G type antiferromagnet, as m increases towards large values, we shall expect that the transition temperature of our superlattices would approach close to the value of 740 K observed in bulk antiferromagnetic LFO. Since we observe a monotonic and rather fast increase of the transition temperature with m , where in samples with three LFO monolayers it is about 170 K, this indicates that the corresponding order is antiferromagnetic. If there was some unusual order in superlattices with few monolayers of LFO, we could expect some non-monotonic dependence on m . However, we agree with the referee that the argument is rather weak and it does not exclude that e.g. the $m = 2$ superlattice exhibits some other order. We have thus rephrased and weakened the corresponding argument in the text.

4. For static and dynamic analysis, I agree with the authors that the longitudinal field measurement could be used to distinguish short-range order and dynamic moment, but the data the author presented is very confusing. The data clearly show that at least there is some static magnetic moment which is also the authors’ argument. However, I fail to understand why the static moment must be from the nuclear moment of SrTiO₃. Is it from Sr, Ti? Can it come from La? Why cannot be from electron spin moment of hybridized O? I would suggest the authors to present the argument more clearly and accessible for general reader since the comparison of static and dynamic is very important in this manuscript.

Response:

We thank the referee for pointing this out. Indeed the argumentation was not clear and actually not necessary for the whole reasoning. For example, the nuclear moments can come from La ions as the referee suggests. We have therefore removed the passage discussing where the nuclear moments are located. This discussion was actually not necessary for the distinguishing between the static and dynamic contributions, which is the main result of the longitudinal field measurement and of the whole paper.

Minor

1. Typo on line 085 should be ‘rare’ instead of ‘rear’

Response:

Thank you, the typo was removed.

Reviewers' Comments:

Reviewer #2:

Remarks to the Author:

In my original review I was criticizing the lack of characterization of the samples which prevent to attribute the observed effect only to dimensionality effects.

The authors wrote in their answers that characterizations of the strain or of the oxygen stoichiometry 'are not probing the magnetic state of the superlattices, which is the topic of this paper and of our conclusion.'

Of course they are not but it is clear that these parameters are strongly influential of the ground state of transition metal oxides and without discussing them how can we trust the conclusions?

The reduced TN of the superlattice remains a concern, and the argument put forward by the authors to address it goes in fact in the wrong direction. If the bulk TN of LFO is reduced by a factor 3 when 50% of La is replaced by Sr, how can they rule out that the reduced TN of the superlattice is simply not due to inter diffusion of Sr on the La sites?

The claims made in the paper are important ones and the authors have to provide evidence supporting them and discarding alternative scenarios. I do not see this in the revised version that I cannot recommend for publication.

Reviewer #3:

Remarks to the Author:

The revised manuscript from Michal Kiaba and other co-authors provides more clearly structural data and some insight regarding the spin fluctuation. In most cases, XRD and RSM provided by the authors are sufficient enough to justify the sample quality from $m=3$ to $m=1$; thus, one can denote the disappearance of AF order to the dimensionality evolution. While the approach to realize a 2D antiferromagnet using oxides superlattice has been shown by several previous work, the result presented by the author is likely one of the closest to what Mermin-Wagner theorem predicts. Overall, I think the revised manuscript shows much improvement and I would recommend the manuscript for publication on Nature Communication.

Dear Editor,

thank you for your reply and suggestion for a revision. We think the question of referee #2 concerning ionic diffusion and related disorder-induced suppression of the magnetic order is important. We summarize our arguments against such a scenario in the reply to the referees. Although we believe we already had such arguments in the previous version of our manuscript, we extended some sections to explain them more concisely.

We hope that these arguments are sufficient for publication in Nature Communications,

best regards, M. Kiaba, and coworkers

Response to referees.

We thank the referees for reviewing our manuscript once more. Notably, we thank Reviewer #3 for recognizing our manuscript's improvements.

response to Reviewer #2

We agree with Referee #2 that ruling out the disorder-induced suppression of the antiferromagnetic order is important. In our manuscript, we present three reasons why this scenario is highly unlikely:

- (i) The strong intensity of the bilayer SL1 and SL-1 XRD peaks with respect to the background in the $m=1$ superlattice demonstrates that the structural boundaries of the bilayer in the superlattices are atomically sharp. These XRD data are incompatible with strong ionic diffusion; therefore, we conclude that the diffusion level is either negligible or low.
- (ii) in the case of $m=2$ superlattice, we observe the static magnetic order below 35K. The latter demonstrates that the structural quality of our superlattices is high enough to preserve the static magnetic order in the $m=2$ case.
- (iii) The strongest argument lies in observing the fluctuating moments in the $m=1$ superlattice at 5K using the longitudinal field measurements. The structural defects like the ionic diffusion would most likely lead to statically disordered magnetic moments. The observation of the fluctuating moment thus makes the disorder-induced scenario of suppression of the antiferromagnetic order highly unlikely and instead points towards a dimensional crossover.

We expanded two sections in the manuscript where we summarized these arguments more concisely.

Reviewers' Comments:

Reviewer #2:

Remarks to the Author:

I believe that the authors has satisfactorily addressed my concerns in the revised version I can now recommend to accept this paper for publication.

Response to referees.

We thank the referees for reviewing our manuscript once more. Below, we give our point-by-point response to your comments. Our replies are highlighted in red color.

REVIEWER COMMENTS

Reviewer #2 (Remarks to the Author):

I believe that the authors has satisfactorily addressed my concerns in the revised version I can now recommend to accept this paper for publication

Reply:

We thank Reviewer #2 for recognizing our manuscript's improvements and for recommending our manuscript for publication.